# Instantaneous antidepressant effect of lateral habenula deep brain stimulation in rats studied with functional MRI

**Gen Li**[1], **Binshi Bo**[2], **Puxin Wang**[1,3], **Peixing Qian**[1,3], **Mingzhe Li**[3], **Yuyan Li**[1], **Chuanjun Tong**[2,4], **Kaiwei Zhang**[2], **Baogui Zhang**[5], **Tianzi Jiang**[5], **Zhifeng Liang**[2]*, **Xiaojie Duan**[1,3,6]*

[1]Department of Biomedical Engineering, College of Future Technology, Peking University, Beijing, China; [2]Institute of Neuroscience, CAS Center for Excellence in Brain Sciences and Intelligence Technology, Key Laboratory of Primate Neurobiology, Chinese Academy of Sciences, Shanghai, China; [3]Academy for Advanced Interdisciplinary Studies, Peking University, Beijing, China; [4]School of Biomedical Engineering, Southern Medical University, Guangzhou, China; [5]Brainnetome Center, Institute of Automation, Chinese Academy of Sciences, Beijing, China; [6]National Biomedical Imaging Center, Peking University, Beijing, China

**\*For correspondence:**
zliang@ion.ac.cn (ZL);
xjduan@pku.edu.cn (XD)

**Competing interest:** The authors declare that no competing interests exist.

**Abstract** The available treatments for depression have substantial limitations, including low response rates and substantial lag time before a response is achieved. We applied deep brain stimulation (DBS) to the lateral habenula (LHb) of two rat models of depression (Wistar Kyoto rats and lipopolysaccharide-treated rats) and observed an immediate (within seconds to minutes) alleviation of depressive-like symptoms with a high-response rate. Simultaneous functional MRI (fMRI) conducted on the same sets of depressive rats used in behavioral tests revealed DBS-induced activation of multiple regions in afferent and efferent circuitry of the LHb. The activation levels of brain regions connected to the medial LHb (M-LHb) were correlated with the extent of behavioral improvements. Rats with more medial stimulation sites in the LHb exhibited greater antidepressant effects than those with more lateral stimulation sites. These results indicated that the antidromic activation of the limbic system and orthodromic activation of the monoaminergic systems connected to the M-LHb played a critical role in the rapid antidepressant effects of LHb-DBS. This study indicates that M-LHb-DBS might act as a valuable, rapid-acting antidepressant therapeutic strategy for treatment-resistant depression and demonstrates the potential of using fMRI activation of specific brain regions as biomarkers to predict and evaluate antidepressant efficacy.

## Editor's evaluation

This important paper is a real tour de force that combines functional MRI, behaviour, and brain stimulation to characterise the effect of stimulation of the lateral habenula in a rodent model for depression. The results are compelling and provide additional information potentially relevant to both surgical targeting and mechanism of action for this deep brain stimulation target.

## Introduction

Major depressive disorder (MDD), also known as depression, is estimated to affect approximately 300 million individuals worldwide and is a leading cause of disability according to the World Health Organization (***World Health Organization, 2017***). Almost 30% of MDD patients fail to respond to

one or more adequate antidepressants and thus exhibit treatment-resistant depression (TRD; *Conway et al., 2017*). TRD is associated with increased morbidity and healthcare costs as well as reduced life quality and work productivity, all of which significantly contribute to the overall burden of MDD (*Mrazek et al., 2014*). Because of the limited effectiveness of available psychological and pharmacological treatments for chronic TRD, various nonpharmacological interventions, including repetitive transcranial magnetic stimulation (rTMS), transcranial direct current stimulation, vagus nerve stimulation, epidural cortical stimulation, electroconvulsive therapy (ECT) and deep brain stimulation (DBS), have been explored as therapeutic options for TRD (*Wong and Licinio, 2001*; *Shelton et al., 2010*; *Cusin and Dougherty, 2012*; *Dandekar et al., 2018*). However, these available treatment choices often have low response rates, multiple (often intolerable) side effects, and substantial lag times before a response is achieved. Lag time is especially dangerous and undesirable because rapid antidepressant effects are critical for patients with suicidal ideation, who account for 15% of TRD patients.

DBS emerged in 2005 and developed into a promising strategy for the management of TRD (*Mayberg et al., 2005*). Clinical and preclinical studies have shown that DBS targeting various brain regions, including the subcallosal cingulate, the ventral capsule/ventral striatum, medial forebrain bundle, nucleus accumbens (NAc), and lateral habenular nucleus (LHb), can induce remission of depressive symptoms (*Anderson et al., 2012*; *Dandekar et al., 2018*; *Zhou et al., 2018*). However, the response rates and effectiveness have varied widely among studies. In addition, most studies have focused on the long-term effects of DBS, on the order of days to weeks; few have investigated the short-term, rapid antidepressant effects (*Wang et al., 2020*; *Scangos et al., 2021*). The mechanisms underlying the therapeutic response to DBS remain unclear. Accumulating evidence indicates that the LHb, which innervates multiple brain regions and directly influences the serotonergic, noradrenergic, and dopaminergic brain systems, exhibits hyperactivity in depressed states (*Hikosaka, 2010*; *Hu et al., 2020*). Recently, LHb-DBS has attracted intense interest for the treatment of TRD. Electrical stimulation of the LHb improved depressive-like behavior in rat models of TRD (*Meng et al., 2011*; *Lim et al., 2015*; *Tchenio et al., 2017*; *Jakobs et al., 2019*). A recent case report observed both short- and long-term improvements in depression, anxiety, and sleep in one human patient after administering high-frequency DBS to the left LHb (*Wang et al., 2020*), suggesting that LHb-DBS may represent a potential form of rapid-acting antidepressant therapy. However, due to the small sample sizes and open label design of relevant studies, the efficacy of this rapid-acting antidepressant therapy and its underlying mechanism remains elusive.

In this study, we observed an immediate (within seconds to minutes) alleviation of depressive-like symptoms with a high-response rate under DBS targeting the LHb (LHb-DBS) in two rat models of depression, Wistar Kyoto (WKY) and lipopolysaccharide (LPS)-treated rats. The remission of depressive symptoms manifested as a significantly increased sucrose preference, decreased immobility time in the forced-swim test, and increased locomotor activity, including increased activity in the center area of an open arena. Simultaneous functional MRI (fMRI) was conducted on the same sets of depressive rats used for behavioral tests under the LHb-DBS (*Figure 1a*). The use of MRI-compatible graphene fiber (GF) electrodes to deliver the DBS pulses enabled complete and unbiased activation pattern mapping across the rat brain by fMRI. LHb-DBS activated multiple regions in afferent and efferent circuitry of the LHb, including those in limbic, serotonergic and dopaminergic systems. The activation levels of brain regions connected to the medial LHb (M-LHb) was correlated with the extent of behavioral improvements. Rats with DBS sites more medial in the LHb exhibited greater antidepressant effects than those with more lateral DBS sites. These results indicate that the antidromic activation of the limbic system and orthodromic activation of the monoaminergic systems connected to the M-LHb play critical roles regarding the instant antidepressant response to LHb-DBS. Our work indicates that DBS of the M-LHb might represent a valuable rapid-acting antidepressant therapy for treating TRD and that fMRI activation of specific brain regions may serve as a biomarker for predicting and evaluating antidepressant efficacy.

## Results

### LHb-DBS instantaneously reduces depressive symptomatology

Bipolar GF microelectrodes were implanted in the right LHb of WKY and LPS-treated rats that showed depressive symptomatology (*Figure 1b–d*). The WKY rat is characterized as an animal model of

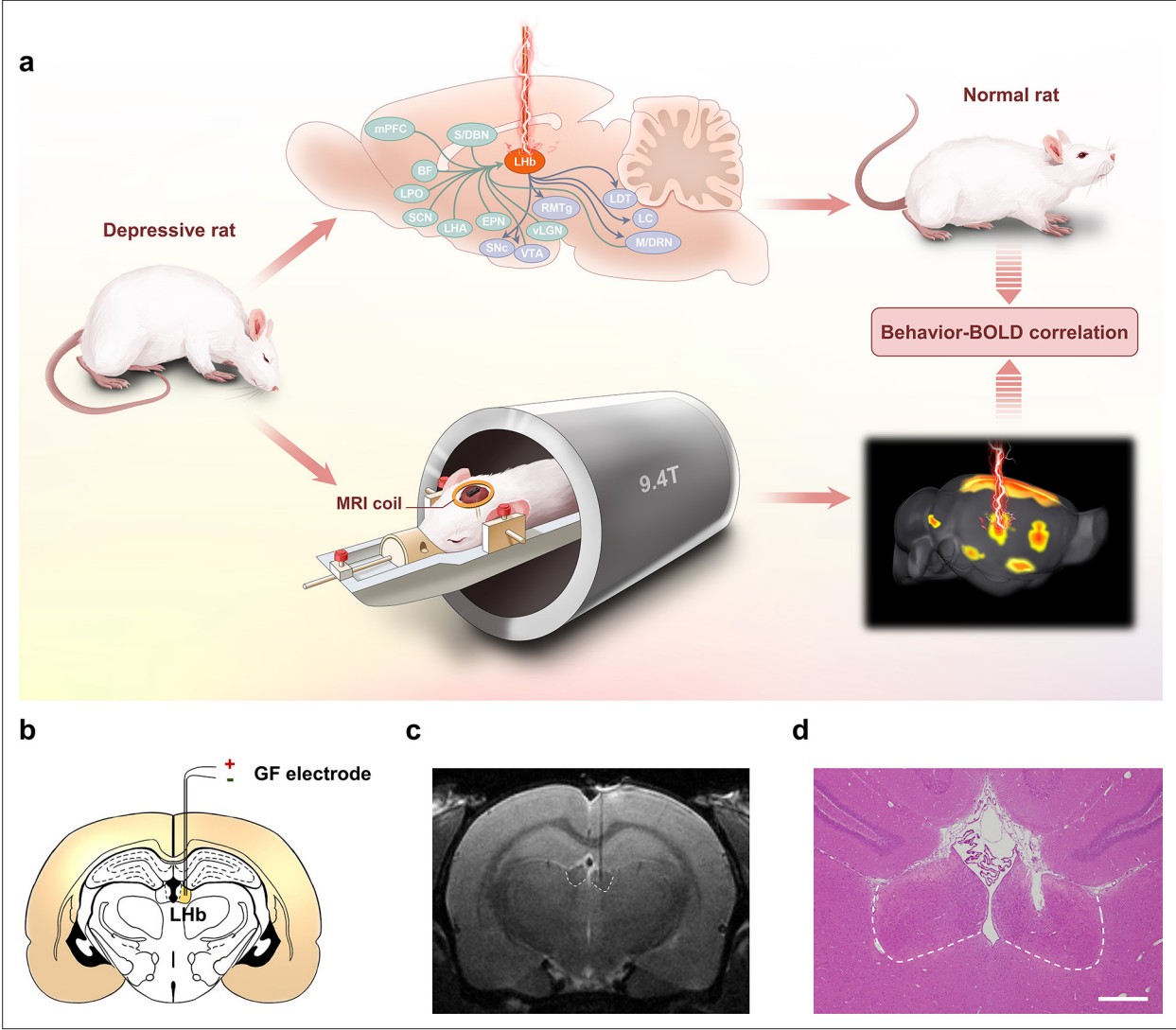

**Figure 1.** Lateral habenula (LHb)-deep brain stimulation (DBS) to rat models of depression studied by functional MRI (fMRI). (**a**) Schematics showing the immediate antidepressant effect of LHb-DBS and fMRI studies in rats. (**b**) A schematic brain section showing the placement of a graphene fiber (GF) bipolar electrode in the LHb of a rat. (**c**) A representative coronal section from the T2-weighted MRI scan of a rat brain with a GF bipolar microelectrode implanted in the LHb. (**d**) An image of a hematoxylin and eosin (H&E) stained brain section with the GF electrode implanted in the LHb from the same rat shown in c. Scale bar, 500 μm.

endogenous depression and exhibits a set of behavioral abnormalities that emulate many symptoms observed in depressive patients, such as increased emotionality and reactivity to stress as well as considerable resistance to classic antidepressants (*Will et al., 2003*; *Aleksandrova et al., 2019*; *Planchez et al., 2019*). As shown in *Figure 2a*, compared to normal Sprague–Dawley (SD, Control) rats, depressive WKY rats exhibited a significantly lower sucrose preference in the sucrose preference test (SPT, 0.49±0.01 versus 0.84±0.02, mean ± SEM, n=10, same for below) and spent more time immobile in the forced swim test (FST, 193±17 s versus 88±14 s). Lower sucrose preference and longer durations of immobility in the FST indicate anhedonia and despair-like behavior, respectively. In the open field test (OFT), which is used to assess anxiety-related behaviors, these WKY rats exhibited decreased locomotor activity, including significantly reduced average speed and increased durations of immobility (*Figure 2a*). In addition, these WKY rats exhibited almost no entries into the center of the open field and spent no time in that area (*Figure 2a*). These results confirmed the presence of depressive symptomatology in WKY rats.

An intraperitoneal injection of LPS is known to increase the levels of inflammatory factors, resulting in depression-like behavior (*Cui et al., 2018*; *Planchez et al., 2019*; *Zhao et al., 2020a*). In our

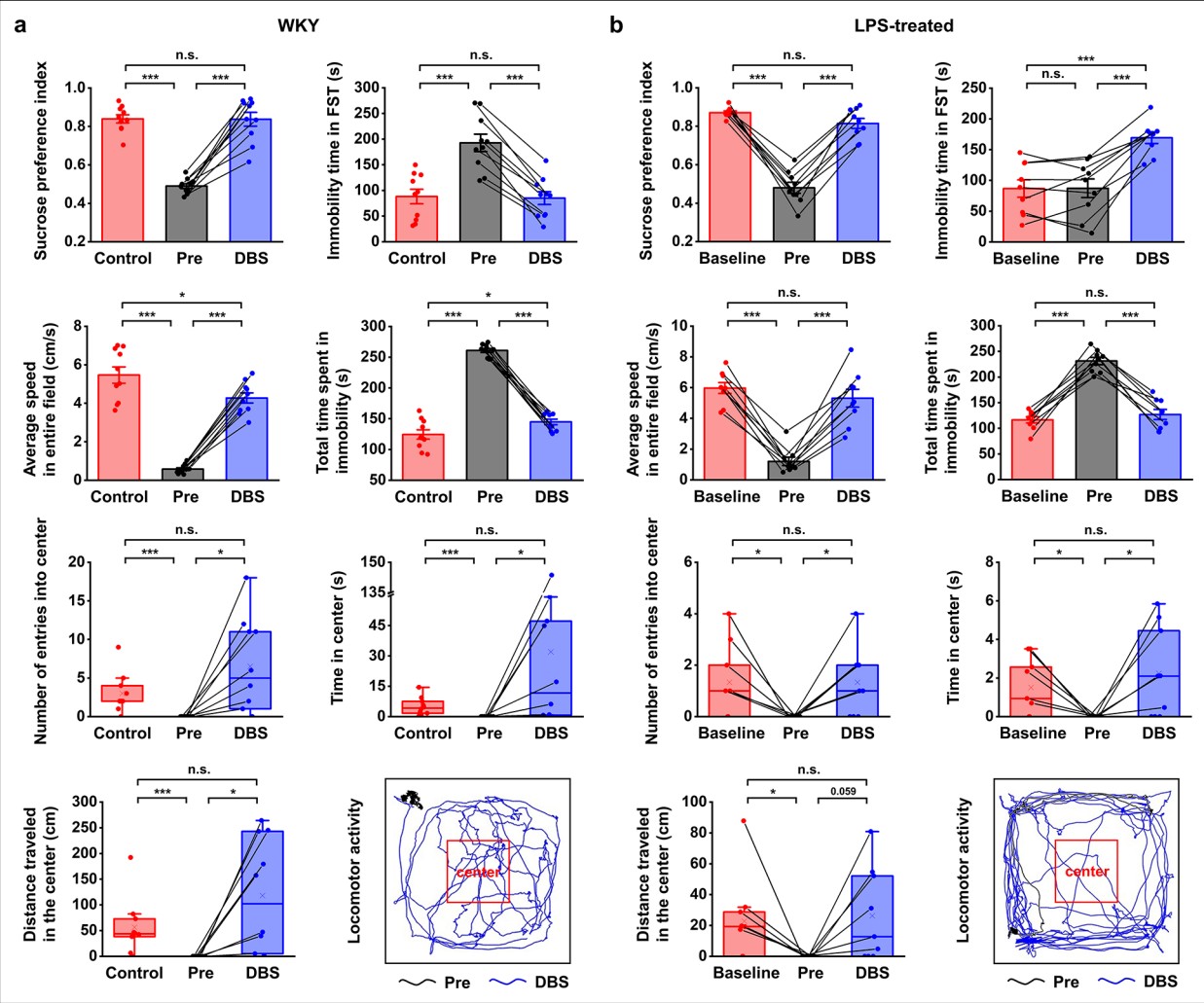

**Figure 2.** Lateral habenula (LHb)-deep brain stimulation (DBS) immediately alleviates depressive-like symptoms in Wistar Kyoto (WKY) and lipopolysaccharide (LPS)-treated rats (models of depression). Quantification of the behavioral responses of WKY (**a**) and LPS-treated (**b**) rats to LHb-DBS (DBS), including the sucrose preference index, the duration of immobility in the forced swim test (FST), average speed in entire field in the open field test (OFT), total duration of immobility in the OFT, number of entries into the center area of the OFT, time spent in the center area of the OFT, and distance traveled in the center area of the OFT. For WKY rats, the behavioral performances of normal Sprague−Dawley (SD) rats (Control) and WKY rats before DBS (Pre) were included for comparison. For LPS-treated rats, the behavioral performances of these rats before LPS injection (Baseline) and before DBS (Pre) were included for comparison. Data from the same animals are connected with black lines. Data are represented as the mean ± SEM (n=10 animals for WKY rats, n=9 animals for LPS-treated rats). The dots indicate data from each individual subject. To compare scores on the sucrose preference test (SPT), FST, and performance in total area of the OFT in WKY rats between the Pre and DBS groups, a two-tailed paired *t* test was used. A two-tailed unpaired *t* test was used for comparisons between the Control and Pre groups as well as between the Control and DBS groups. Because the data on variables from the center area of the OFT were not normally distributed in WKY rats, Wilcoxon's matched pairs signed-rank test was used to compare the Pre and DBS groups, and the Mann−Whitney *U* test was used to compare the Control and Pre groups as well as the Control and DBS groups. Data from the SPT, FST, and total area of the OFT in LPS-treated rats were compared by one-way repeated-measures ANOVA tests with Tukey post hoc tests. Because the data from the center area of the OFT of LPS-treated rats was nonnormally distributed, Friedman's tests were used for comparison. n.s.: not significant, *p<0.05, **p<0.01, and ***p<0.001. The last panels in (**a**) and (**b**) show examples of the locomotor activity of a WKY rat and an LPS-treated rat before (Pre, black line, 5 min) and during (DBS, blue line, 5 min) LHb-DBS with graphene fiber (GF) bipolar electrodes. The area outlined by the red square was defined as the center area, accounting for one-ninth of the total area size. See *Figure 2—figure supplements 1 and 2* for additional details. Results from detailed statistical tests are summarized in ***Supplementary file 1***.

The online version of this article includes the following figure supplement(s) for figure 2:

**Figure supplement 1.** Additional behavioral response indicators in open field test (OFT) of Wistar Kyoto (WKY) and lipopolysaccharide (LPS)-treated rat models of depression.

**Figure supplement 2.** Behavior and functional MRI (fMRI) activation patterns of depressive rat models under deep brain stimulation (DBS) with stimulating graphene fiber (GF) electrodes implanted outside lateral habenula (LHb).

experiments, LPS-treated Wistar rats exhibited a significant reduction in sucrose preference and locomotor activity, including decreased average speed and increased immobility time in the OFT (*Figure 2b*, n=9 rats). The numbers of entries, time spent within, and total distance traveled in the center area of the OFT were also significantly reduced (*Figure 2b*, n=9 rats). No significant change was observed in the immobility time in the FST (*Figure 2b*, n=9 rats). This finding is consistent with previous observations that LPS administration at lower doses did not alter behavioral performance in the FST (*Tonelli et al., 2008*; *Pitychoutis et al., 2009*). Taken together, these results indicate the successful establishment of an inflammatory model of depression, given the clear anhedonia and anxiety-like behaviors.

Electrode tip placements within the LHb for DBS were verified for each subject by T2-weighted rapid acquisition with relaxation enhancement (RARE) anatomical MRI acquired immediately after implantation (*Figure 1c*). The negligible artifact induced by the GF electrodes did not obscure the LHb and allowed accurate identification of the electrode tip positions in MRI scans. These advantages enabled simple and precise in vivo verification of the placement of the implanted electrodes. Electrode tip localization within the LHb was also confirmed by hematoxylin and eosin (H&E) staining at the end of the study which showed consistent results as those from MRI scans (*Figure 1d*). Unless otherwise specified, rats with GF electrodes successfully implanted into the LHb were selected and used in the behavioral tests and DBS-fMRI experiments.

High-frequency stimulation consisting of 130 Hz constant-current pulses with an amplitude of 300 µA and a duration of 90 µs (biphasic and symmetric) was applied to the GF bipolar microelectrodes implanted in rat models of depression. After 15 min of LHb-DBS, the sucrose preference index (SPI) of the WKY rats increased from 0.49±0.01 to 0.84±0.04 (mean ± SEM, n=10, same for below; *Figure 2a*). This immediate improvement in sucrose preference indicates the rapid and significant reversal of anhedonia in these WKY rats by LHb-DBS. The despair-like symptoms were also instantly alleviated upon the administration of LHb-DBS, as indicated by the decreased immobility time in the FST from 193±17 s to 85±12 s (*Figure 2a*). In the OFT, LHb-DBS significantly increased the average speed from 0.58±0.07 cm s⁻¹ to 4.27±0.26 cm s⁻¹ and decreased the duration of immobility from 260.97±3.24 s to 144.85±4.49 s. In addition, upon LHb-DBS, these rats made more entries into (0 versus 6.50±1.96), spent more time in (0 s versus 32.00±14.33 s), and traveled a longer distance (0 cm versus 118.08±35.01 cm) in the center area. An example of the locomotor activity of a WKY rat before (black line, 5 min) and during (blue line, 5 min) LHb-DBS is shown in *Figure 2a*. Together, these results indicated that LHb-DBS significantly alleviated depressive symptomatology in WKY rats. More importantly, this antidepressant effect occurred immediately or within a few minutes after the administration of LHb-DBS, indicating a short lag time to response.

The immediate antidepressant effect of LHb-DBS was also observed in LPS-treated depressive rats. Upon administration of LHb-DBS, LPS-treated depressive rats showed a significant increase in sucrose preference and increased locomotor activity, including behavior in the center area of the OFT. Specifically, the SPI increased from 0.48±0.03 to 0.81±0.03 (mean ± SEM, n=9 rats, same for below). In the OFT, the average speed increased from 1.21±0.27 cm s⁻¹ to 5.31±0.58 cm s⁻¹, the duration of immobility decreased from 231.21±7.14 s to 126.91±9.81 s, the number of entries into the center area increased from 0 to 1.33±0.44, the time spent in the center increased from 0 s to 2.24±0.79 s, and the total distance traveled in the center increased from 0 cm to 26.25±9.99 cm (*Figure 2b*). The increases in locomotor activity were also reflected by the increased total distance traveled (in the entire field) as well as the increased average speed in the center area (*Figure 2—figure supplement 1*). An example of the locomotor activity of an LPS-treated rat before (black line, 5 min) and during (blue line, 5 min) LHb-DBS is shown in *Figure 2b*. Due to the lack of behavioral despair in LPS-treated rats, we did not observe any decrease in the duration of immobility in the FST in these rats upon the administration of LHb-DBS. Instead, we observed an increased duration of immobility. This increase is consistent with the time effect of the FST reported in a previous minute-by-minute analysis of the FST (*Pitychoutis et al., 2009*; *Mezadri et al., 2011*; *Costa et al., 2013*).

Overall, WKY and LPS-treated depressive rats showed a high-response rate to LHb-DBS. Of 19 WKY and LPS-treated rats, 16 (~84.2%) showed an increase in sugar preference greater than 50%, 13 (~68.4%) showed an increase in average speed greater than fivefold, and 14 (~73.7%) exhibited at least one entry into the center area in the OFT compared to no entries before DBS. Of 10 WKY rats, 6 had a decrease in the duration of immobility in the FST greater than 50%. In addition, DBS of same

parameters delivered to electrodes implanted outside the LHb failed to exert antidepressant effects (*Figure 2—figure supplement 2a*).

## fMRI studies of LHb-DBS

The fMRI scans were performed during LHb-DBS on the same sets of WKY and LPS-treated rat models of depression under anesthesia. The stimulation pulses were the same as those used for behavioral tests, except that a higher pulse amplitude of 600 µA was used to achieve a more robust effect and circumvent the issue of low DBS-fMRI sensitivity at low stimulus amplitude (*Albaugh et al., 2016*). The 600 µA stimulation generated blood-oxygenation-level-dependent (BOLD) signal patterns qualitatively similar to but more robust than stimulation at 300 µA. Notably, the high-charge injection capacity of the GF electrodes allowed for the delivery of the 600 µA pulses without polarizing the electrode beyond the potentials for water reduction or oxidation with a small electrode size of only ~75 µm, thus ensuring both safety and stimulation resolution (*Zhao et al., 2020b*).

The small-to-absent artifact produced by the GF electrodes enabled fMRI scanning of all brain regions, thus resulting in full and unbiased activation pattern mapping under LHb-DBS in rat models. Robust positive BOLD responses were evoked ipsilaterally in multiple regions along the direct afferent and efferent circuitry of the LHb as well as several regions outside the direct circuitry and the DBS target (the LHb). Representative BOLD activation maps of a WKY rat and a LPS-treated rat are shown in *Figure 3b and c*, and individual activation maps from all the WKY rats and LPS-treated rats used in this study are shown in *Figure 3—figure supplements 1 and 2*. The activation patterns were similar in all rats, although individual rats differed in the intensity of BOLD activation. The activated brain regions with afferent connections to the LHb included the septum, diagonal band nucleus (DBN), lateral preoptic area (LPO), lateral hypothalamic area (LHA), and medial prefrontal cortex (mPFC). The ventral tegmental area (VTA) and dorsal raphe nucleus (DRN) are two activated regions with efferent connections from the LHb. Since the interpeduncular nucleus (IPN) is close to the VTA and it is difficult to accurately distinguish these two areas in fMRI studies, we labeled this region as IPN/VTA. In addition, we observed the activation of the sublenticular extended amygdala (SLEA), cingulate cortex (Cg), and retrosplenial cortex (RS), which are not directly connected to the LHb. Despite DBS-induced improvements in locomotor activity, no significant BOLD activation in the motor cortex was observed under LHb-DBS.

The time courses of the BOLD signals in several anatomical regions of interest (ROIs) were calculated and averaged from all WKY and LPS-treated rats. We observed clear BOLD signal changes time-locked to the stimulation pulse blocks (*Figure 3d*). Of all of the regions examined in WKY rats, the mPFC showed the largest percent changes in BOLD signals (4.87 ± 0.42%, mean ± SEM, n=30 scans from 10 rats), slightly higher than those in the DBS target, the LHb (4.77 ± 0.38%, mean ± SEM, n=30 scans from 10 rats). For LPS-treated rats, the LHb exhibited the largest changes in BOLD signals (7.02 ± 0.35%, mean ± SEM, n=27 scans from 9 rats), and the mPFC exhibited the second largest changes (4.79 ± 0.28%, mean ± SEM, n=27 scans from 9 rats). The limbic areas, including the septum, DBN, LPO, and LHA, exhibited an overall lower BOLD response than the cortical regions, including the mPFC, Cg, and RS. The IPN/VTA showed activation approximating that of the cortical regions, and the activity level of the DRN was close to that of limbic regions. A characteristic "double peak" in the BOLD signal was observed in specific regions, including the Cg, mPFC, and DRN, possibly due to the recruitment of two distinct circuitries or a delayed neurotransmission effect (*Van Den Berge et al., 2017*). The BOLD signals of those depressive rats with electrodes implanted outside the LHb showed distinctly different patterns from those with accurate electrode implantation into the LHb. Some rats showed almost no activation across the entire brain, and others showed activation only in the target LHb, Cg, and mPFC (*Figure 2—figure supplement 2b*).

## Correlation between antidepressant efficacy and fMRI responses

To shed light on the mechanism underlying the immediate antidepressant effect of LHb-DBS, we conducted Pearson correlation analysis on the behavioral improvement and the BOLD activation levels of various brain regions in individual rats. The increase in average speed in the OFT, as characterized by the ratio of speed with and without DBS, the number of entries into the center of the open field, and the distance traveled in the center were significantly correlated with the beta values of BOLD signals in the septum, DBN, LPO, LHA, DRN, and SLEA (p<0.05, *Figure 4a and b*, *Figure 4—figure*

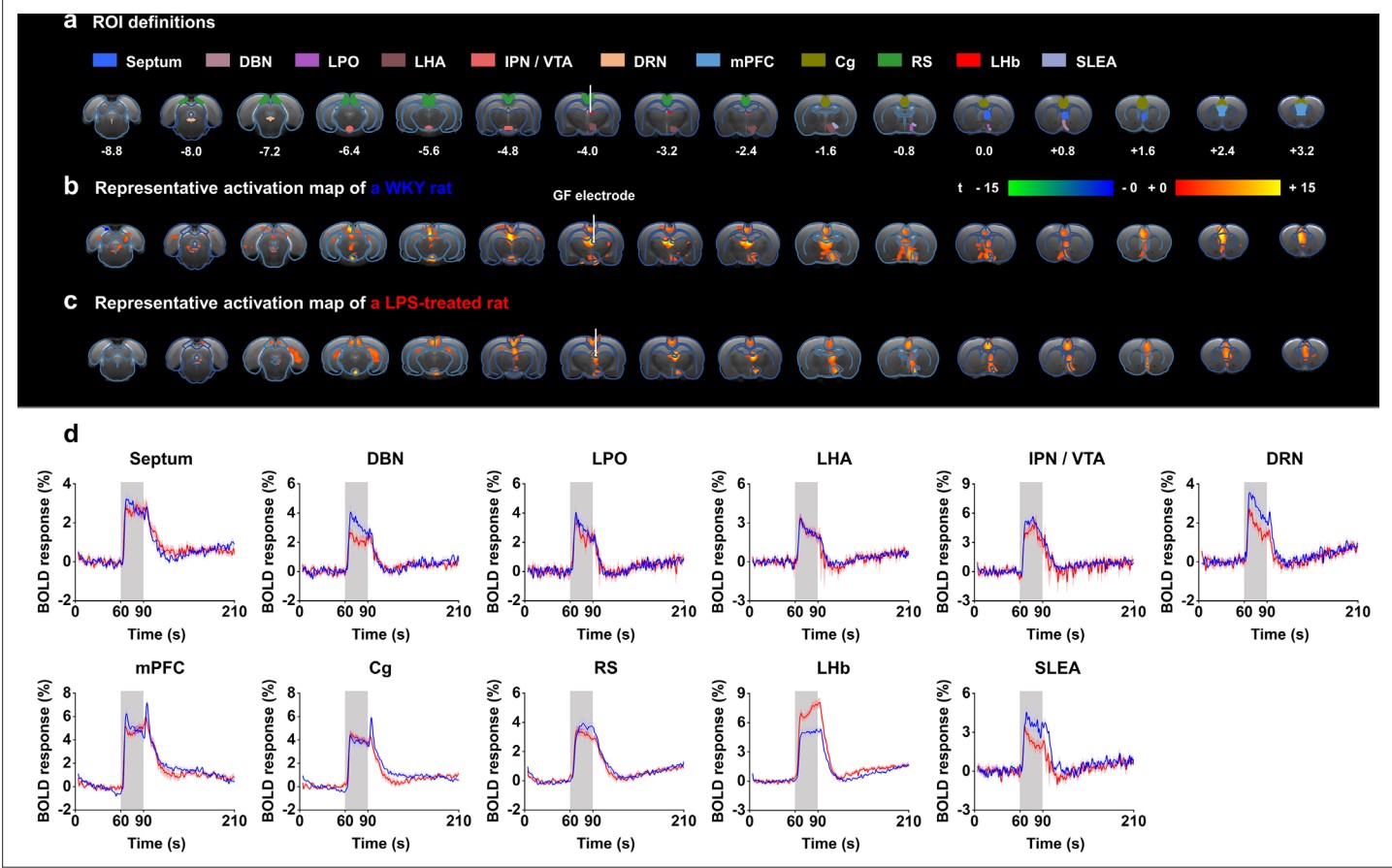

**Figure 3.** Functional blood-oxygenation-level-dependent (BOLD) activation evoked by lateral habenula (LHb)-deep brain stimulation (DBS) with graphene fiber (GF) stimulating electrodes in rat models of depression. The same sets of rats were used as those in the behavioral tests. (**a**) Definition of regions of interest (ROIs) for different brain regions. The numbers below slices denote the relative distance from bregma (in mm). The same set of distance numbers applies to the slices in **b and c**. (**b and c**) Representative BOLD maps of a Wistar Kyoto (WKY) rat (**b**) and a lipopolysaccharide (LPS)-treated rat (**c**). The BOLD activation maps are overlaid onto averaged anatomical images. The color bar denotes t-score values obtained by GLM analyses, with a significance threshold of corrected p<0.001. The vertical white lines in a–c indicate the graphene fiber (GF) bipolar electrode. (**d**) BOLD signal time series at anatomically defined ROIs evoked by LHb-DBS in WKY (blue) and LPS-treated (red) rats. The stimulation epoch is indicated by the gray-shaded band. The solid lines show the average signal, and the shaded regions represent the SEM, n=30 scans from 10 WKY, n=27 scans from 9 LPS-treated rats. DBN, diagonal band nucleus; LPO, lateral preoptic area; LHA, lateral hypothalamic area; IPN, interpeduncular nucleus; VTA, ventral tegmental area; DRN, dorsal raphe nucleus; mPFC, medial prefrontal cortex; Cg, cingulate cortex; RS, retrosplenial cortex; LHb, lateral habenula; SLEA, sublenticular extended amygdala. See *Figure 3—figure supplements 1 and 2* for additional details.

The online version of this article includes the following figure supplement(s) for figure 3:

**Figure supplement 1.** Individual blood-oxygenation-level-dependent (BOLD) activation maps evoked by lateral habenula (LHb)-deep brain stimulation (DBS) from the 10 Wistar Kyoto (WKY) depressive rats.

**Figure supplement 2.** Individual blood-oxygenation-level-dependent (BOLD) activation maps evoked by lateral habenula (LHb)-deep brain stimulation (DBS) from the 9 lipopolysaccharide (LPS)-treated depressive rats.

supplement 1) but not correlated with those in the Cg, RS, mPFC, and LHb (*Figure 4—source data 1*). The activation of the IPN/VTA showed a relatively weak correlation with the increase in average speed, but no correlation was observed between that and the number of entries into the center or distance traveled in the center (*Figure 4a and b*, *Figure 4—source data 1*). The activation intensity of the DRN showed the strongest correlation with the increase in the average speed ($r=0.92$, $p=2.3 e^{-8}$) among all the brain regions and was also significantly correlated with the number of entries into the center ($r=0.60$, $p=0.007$) and the distance traveled in the center ($r=0.55$, $p=0.01$; *Figure 4a and b*, *Figure 4—figure supplement 1*). Notably, except for the SLEA, all the brain regions with activation levels correlated with the above three indicators of the OFT have direct afferent or efferent

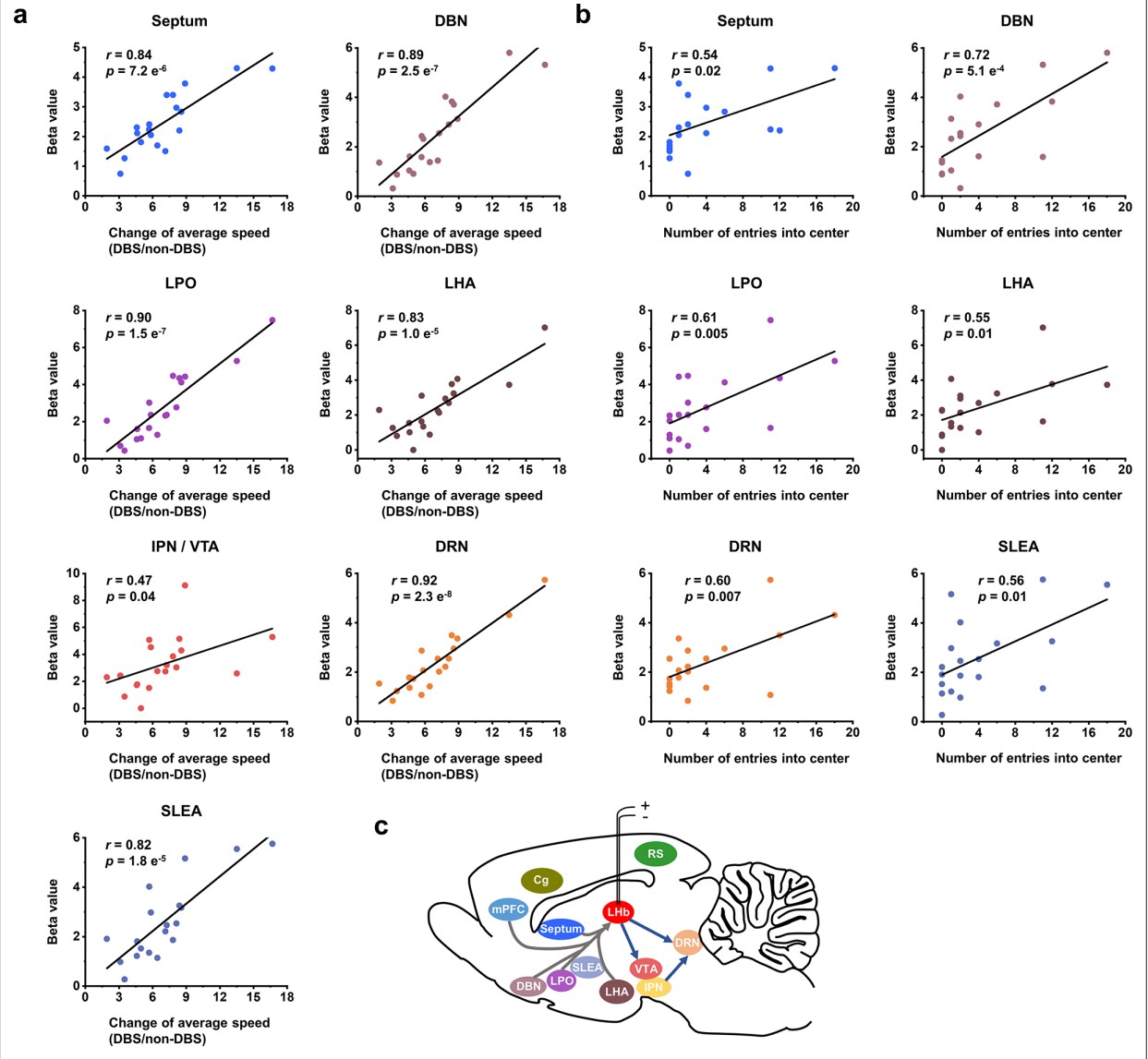

**Figure 4.** Correlation between functional MRI (fMRI) responses and behavioral improvement. Scatter plots of the regional beta values of blood-oxygenation-level-dependent (BOLD) responses and performance indicators on the open field test (OFT) from all the Wistar Kyoto (WKY) and lipopolysaccharide (LPS)-treated rats upon administration of lateral habenula (LHb)-deep brain stimulation (DBS), including (**a**) change of average speed, defined as the ratio of average speed with DBS (DBS) and without DBS (non-DBS), and (**b**) number of entries into the center. The Pearson's correlation coefficient *r* between behavioral improvement in the OFT and beta values of BOLD responses across rats were calculated for each regions of interest (ROI). (**c**) A schematic showing the placement of the stimulating electrode in LHb and summary of the activated brain areas. Arrows to and from LHb indicate afferent and efferent connections, respectively. See *Figure 4—figure supplement 1* and *Figure 4—source data 1* for additional details.

The online version of this article includes the following source data and figure supplement(s) for figure 4:

**Figure supplement 1.** Correlation between functional MRI (fMRI) responses and distance traveled in center.

**Source data 1.** Correlation analysis results between functional MRI (fMRI) responses in some brain areas and behavioral improvement.

connections to the LHb (*Figure 4c*). No correlation was observed between the BOLD activation levels and sucrose preference or the duration of immobility in the FST.

## Antidepressant efficacy depended on DBS position

The excellent MRI compatibility of the GF electrodes facilitated exact determination of the locations of electrode tips within the LHb in vivo, thus allowing us to correlate the antidepressant efficacy with the stimulating positions relative to the anatomical substructures of the target nuclei. On the basis of

differential afferent and efferent connections, the LHb can be divided into the M-LHb and the lateral LHb (L-LHb; *Hu et al., 2020*). Based on the T2-weighted anatomical images, we delineated the electrode tip location of each rat and summarized them in *Figure 5a*. T2-weighted MRI images of all the WKY rats and LPS-treated rats used in this study are included in *Figure 5—figure supplement 1*.

The rats with DBS sites in the M-LHb and close to the medial habenula (MHb), as indicated by the red dots in *Figure 5a*, showed greater improvements in average speed than the rats with more lateral stimulation sites, especially those in the L-LHb and far from the MHb, as indicated by the blue dots (*Figure 5b*). The behavioral performance in the center area, including the number of entries into the center, time spent in the center, and distance traveled in the center, also exhibited greater improvement in the rats with DBS sites in the M-LHb and close to MHb compared to those in L-LHb and far from MHb, with the difference approaching the borderline of significance (*Figure 5b*). This stimulation position-dependent differences are also shown in the scatter plots of electrode tip positions and behavioral response indicators (*Figure 5—figure supplement 2*). Consistent with these findings, the BOLD activation levels of brain regions with activation level correlated with behavioral improvements, including the septum, DBN, LPO, LHA, IPN/VTA, DRN, and SLEA, also showed clear position-dependent differences. Rats with more medial DBS sites had stronger BOLD activation in these regions than rats with more lateral DBS sites (*Figure 5b*). This dependence of BOLD activation on stimulation position is clearly indicated by the difference in BOLD activation maps averaged from the rats of the three groups, respectively (*Figure 5c*).

## Discussion

We observed instantaneous alleviation of depressive-like symptoms with a high-response rate in WKY and LPS-treated rats induced by LHb-DBS with GF electrodes. Low-response rates and substantial lag times before response are the main limitations of available psychological and pharmacological treatments for TRD (*Wong and Licinio, 2001*; *Shelton et al., 2010*). These long lag times are dangerous and undesirable because it is associated with a high risk of suicidal behavior (*Jick et al., 2004*). ECT produced a prompt improvement in symptoms of depression in the majority of patients treated (*Lisanby, 2007*). However, the relatively higher risk of side effects, such as cognitive impairment, limits the application of this treatment. Ketamine administration also showed a rapid effect (within hours) in patients with TRD, and this effect lasted up to 1–2 weeks. However, there are issues surrounding the long-term efficacy and safety of repeated administration of ketamine (*Corriger and Pickering, 2019*; *Krystal et al., 2019*). Clinical studies on the use of DBS to treat TRD have demonstrated clinically relevant antidepressant effects (*Dandekar et al., 2018*; *Zhou et al., 2018*). However, early results were not consistently replicated across studies. In addition, most clinical and preclinical studies on the use of DBS to treat TRD using various stimulation targets, including the subcallosal cingulate, the ventral capsule/ventral striatum, medial forebrain bundle, and NAc, focused on its long-term effect, which requires days to weeks of stimulation before a response is achieved (*Schlaepfer et al., 2014*). The rapid-acting antidepressant effect of DBS has rarely been reported (*Wang et al., 2020*; *Scangos et al., 2021*).

In our studies, we found that out of the 19 WKY and LPS-treated rats, 16 (~84.2%) exhibited a sucrose preference increase greater than 50%, 13 (~68.4%) showed an increase in average speed greater than fivefold, and 14 (~73.7%) had at least one entry into the center area in the OFT. Of the 10 WKY rats, 6 experienced a decrease in the duration of immobility in the FST greater than 50%. More importantly, the depressive behavior was alleviated immediately upon delivery of the stimulation pulses or within a few minutes. The WKY and LPS-treated depressive rat models share similar characteristics, including abnormalities in various neurotransmitter and endocrine systems and emotional changes resulting from inflammatory stimuli. These models are widely used in pharmacological and nonpharmacological depression treatment studies (*Caldarone et al., 2015*; *Aleksandrova et al., 2019*; *Lasselin et al., 2020*). Previous research indicates that classic antidepressants used in humans, such as selective serotonin reuptake inhibitors, also cause an antidepressant reaction in WKY rats. Ketamine, a rapid-acting antidepressant in clinical practice, has been shown to be effective in both WKY and LPS-treated rats (*Aleksandrova et al., 2019*; *Zhao et al., 2020a*). In WKY rats, DBS of the NAc increased exploratory activity and exerted anxiolytic effects, and NAc-DBS was found to be effective for TRD treatment in humans (*Dandekar et al., 2018*; *Aleksandrova et al., 2019*). These results suggest that the depression rat models can provide valuable information about the efficacy

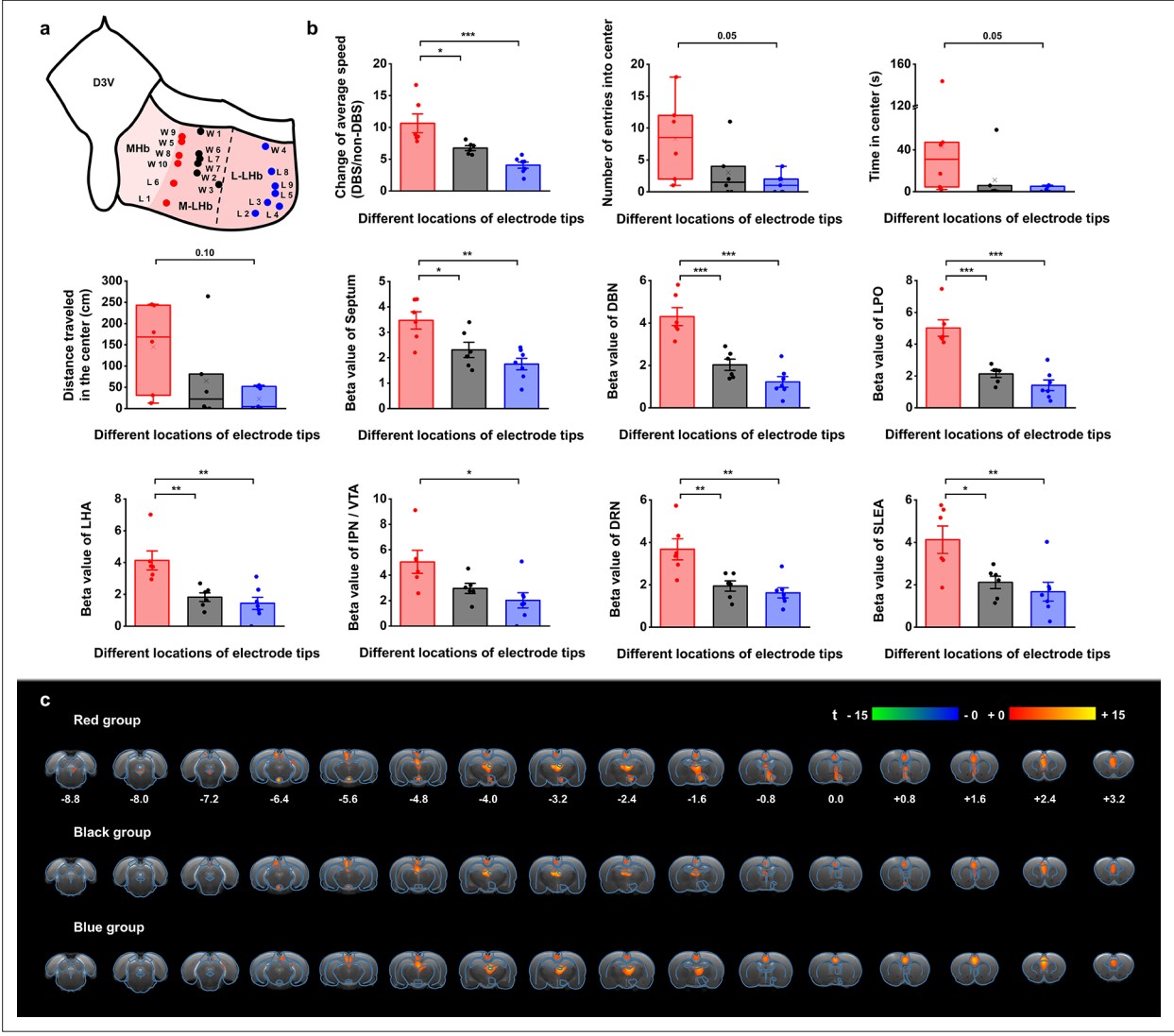

**Figure 5.** Stimulation position dependence of the antidepressant efficacy and blood-oxygenation-level-dependent (BOLD) responses. (**a**) A schematic diagram showing the positions of the graphene fiber (GF) electrode tips within the habenula from all the Wistar Kyoto (WKY) and lipopolysaccharide (LPS)-treated depressive rats. Each dot represents the electrode tip position in a depressive rat. Labels with "W" indicate WKY rats and those with "L" indicate LPS-treated rats. According to the positions in the medial-lateral direction, the depressive rats were divided into three groups, with red dots corresponding to the group with stimulation positions more medial, black dots corresponding to the group with stimulation positions in the middle, and blue dots corresponding to the group with stimulation positions more lateral in lateral habenula (LHb). The electrode tip position was delineated from the T2-weighted MRI images of each rat. The dashed line indicates the boundary between the medial LHb (M-LHb) and the lateral LHb (L-LHb). (**b**) Performance indicators in the open field test (OFT) and beta values of BOLD responses in several regions of interest (ROIs) from the three groups of rats with different stimulation locations. The change of average speed is defined as the ratio of average speed with deep brain stimulation (DBS) and without DBS (non-DBS). The number of entries into the center, time spent in the center and distance traveled in the center area of the OFT correspond to the behavioral performance with DBS. *p<0.05, **p<0.01, ***p<0.001, one-way ANOVA tests with Tukey post hoc analysis for change of average speed, Kruskall–Wallis ANOVA for indicators in the center of the OFT, one-way ANOVA tests with Tukey post hoc analysis for beta values of BOLD responses. Red, black, and blue histogram bars represent the three groups marked with red, black, and blue dots in (**a**). The dots represent the data of each individual rat. The error bars indicate the SEM. (**c**) BOLD activation maps averaged from the three groups of rats with stimulation positions marked with red, black, and blue dots in (**a**) respectively. The numbers below slices denote the relative distance from bregma (in mm). The color bar denotes t-score values obtained by GLM analyses, with a significance threshold of corrected p<0.001. See *Figure 5—figure supplements 1 and 2* for additional details. Results from detailed statistical tests are summarized in *Supplementary file 1*.

The online version of this article includes the following figure supplement(s) for figure 5:

**Figure supplement 1.** Individual coronal T2-weighted MRI images from all the Wistar Kyoto (WKY) and lipopolysaccharide (LPS)-treated depressive rats used in behavioral tests and functional MRI (fMRI) studies.

**Figure supplement 2.** Color-coded scatter plots of medial-lateral (ML) positions of electrode tips and different behavioral response indicators.

of various pharmacological and nonpharmacological therapies. In a recent case report, researchers observed acute stimulation effects in addition to long-term clinical improvements in depression, anxiety, and sleep in a patient with TRD upon administering LHb-DBS (*Wang et al., 2020*). This finding supports the clinical relevance of our observations. However, no animal model of depression can completely replicate human symptoms, and further research is necessary to validate our findings in human patients. Additionally, the long-term efficacy and side effects of LHb-DBS require further investigation. Nevertheless, we believe that our findings propose a promising addition to the rapid-acting therapeutic options for the most refractory depression patients.

Simultaneous fMRI scans during LHb-DBS revealed the activation of multiple regions in afferent and efferent circuitry of the LHb. The LHb receives afferent inputs mainly from the limbic forebrain regions and the basal ganglia, which funnel through largely parallel circuits to reach the M-LHb and L-LHb, respectively (*Hu et al., 2020*). No obvious activation was observed in basal ganglia regions afferent to the L-LHb. Except for the mPFC, which is an afferent connection of the L-LHb, all the activated afferent regions were in the limbic areas connected to the M-LHb. The activation levels of these limbic areas showed a strong correlation with the extent of the alleviation of depressive symptoms. However, this correlation was not observed in the mPFC. The VTA and DRN are regions efferent from the LHb activated by LHb-DBS; these regions are also connected to the M-LHb. The activation levels of these two regions, especially that of the DRN, were also correlated with the antidepressant efficacy. The rostromedial tegmental nucleus, which receives the majority of LHb output and is efferently connected to the L-LHb, was not activated. All these results indicate that the immediate antidepressant effect of DBS mainly originated from the stimulation of the M-LHb and the antidromic and orthodromic activation of the circuits connected to the M-LHb, including the limbic, serotonergic and dopaminergic systems. This theory is further supported by the findings that stimulation in more medial regions of the LHb produced more potent antidepressant effects than stimulation in more lateral regions. In addition, previous studies found that bursts of neuronal firing, an electrophysiological characteristic related to depression, occurred preferentially in the M-LHb instead of the L-LHb (*Yang et al., 2018*). We believe that these findings provide important guidance in the development and optimization of LHb-DBS therapy for TRD. For example, considering the small size of LHb, stimulation should be confined to the M-LHb to maximize the antidepressant efficacy and minimize the associated side effects by utilizing high-resolution stimulation electrodes and parameters. Furthermore, the correlation between behavioral improvement and BOLD activation levels in specific regions provides a potential biomarker in addition to describing individual clinical characteristics for predicting and evaluating the treatment response, which is important for the emerging framework of precision psychiatry.

In addition to the brain regions with direct afferent or efferent connections to the LHb, we observed the activation of the SLEA, Cg, and RS regions that are not implicated in the LHb circuits by LHb-DBS. Of these areas, only the SLEA showed an activation level that were correlated with the extent of alleviation of depressive symptoms. The SLEA is a region with multiple connections to other limbic brain regions and potentially capable of coordinating activity in different areas of the limbic lobe forebrain involved in emotional processing (*Abler et al., 2007*). We suspect that the activation of the SLEA by LHb-DBS was mediated by the activation of limbic brain structures. Previous studies on humans have shown that the Cg and RS have resting-state functional connections with the habenula (*Ely et al., 2016*; *Torrisi et al., 2017*). This connection was associated with subclinical depression scores (*Ely et al., 2016*). Functional connections with the habenula may account for the activation of the Cg and Rs. Although the Cg has been investigated as a potential predictive biomarker of treatment response in depression across diverse treatment modalities, including antidepressant medications, evidence-based psychotherapy, rTMS, and ECT (*Ge et al., 2020*), we did not observe any correlation of Cg activation level with antidepressant efficacy in the present study. This suggests that the activation of the LHb-Cg/RS pathway did not play a critical role in the immediate antidepressant effect observed here.

Upon administration of LHb-DBS, both WKY and LPS-treated rats showed significant improvement in locomotor activity. However, no obvious activation by LHb-DBS was observed in the motor cortex for either type of rat. It was reported that the suppression of motor action in depressive rats arises from the inhibition of dopaminergic and serotonergic neurons in the VTA, substantia nigra pars compacta and raphe nuclei due to hyperactivity of the LHb (*Hikosaka, 2010*). The absence of motor cortex activation by LHb-DBS suggests that the improvement of the locomotor activity was not due to the direct activation of the motor circuit. Instead, this improvement might have resulted

from the reversal of motor suppression by activating the limbic and monoaminergic systems, such as the DRN and VTA. Mounting evidence from animal and human studies indicates that hyperactivity of the LHb plays a critical role in depression etiology (*Hikosaka, 2010*; *Hu et al., 2020*). DBS targeting the LHb might result in an immediate suppression of hyperactivity and modulate the transmission of monoamine neurotransmitters, such as dopamine and serotonin, in interconnected brain areas, thus resulting in the immediate remission of depressive symptoms, including anhedonia, despair, and anxiety. Notably, the DRN, as the main source of serotonin in the brain, exhibited the strongest correlation in activation level with behavioral improvement among all the brain regions. The M-LHb sends direct glutamatergic projections to both serotonergic neurons and GABAergic neurons in the DRN, which in turn exert feedforward inhibition of serotonergic neurons in the raphe nucleus (*Hu et al., 2020*). The strong correlation between the activation level and behavioral improvement in the DRN suggests that the reversal of the inhibition of DRN serotonergic neurons might play a critical role in the rapid antidepressant effect of LHb-DBS.

Notably, the high-MRI compatibility of the GF electrodes used for delivering the stimulation played an important role in our fMRI evaluation of the effects of LHb-DBS. Previous DBS-fMRI studies of other brain targets for depression, such as the NAc, have failed to demonstrate reliable BOLD responses within the DBS target and brain areas near the electrode tracks due to the severe artifact produced by conventional stimulating electrodes (*Knight et al., 2013*; *Albaugh et al., 2016*). In addition to allowing in vivo localization of DBS sites within the LHb, the small-to-absent artifact produced by the GF electrodes used here enabled full and unbiased mapping of fMRI activation patterns in all brain regions, which is important for understanding the therapeutic mechanisms underlying DBS. Furthermore, we conducted fMRI studies on the same sets of animals used for behavioral tests. This paradigm is critical for revealing the neuromodulatory effects and underlying mechanism of DBS.

In conclusion, an instant alleviation of depressive-like symptoms with a high-response rate was observed in WKY and LPS-treated depressive rats upon administration of LHb-DBS. DBS-fMRI scans revealed that this antidepressant efficacy was correlated with activation levels of multiple regions in the limbic and monoaminergic systems afferently or efferently connected to the M-LHb. Stimulation in more medial subregions of the LHb produced more potent antidepressant effects than stimulation in more lateral subregions. The findings here provide important insights for the development of future DBS therapy for refractory depression with improved outcome, including short response lag time, high-response rate, and high efficacy.

## Methods

### Animals

Our procedures for handling the animals complied with the Beijing Administration Rules of Laboratory Animals and the National Standards of Laboratory Animal Requirements of Environment and Housing Facilities (GB 14925–2010) and were approved by the Institutional Animal Care and Use Committee of Peking University (#FT-DuanXJ-1). Male WKY rats, SD rats, and Wistar rats 8–12 weeks old (Charles River Laboratories, China) were used throughout this study. Wistar rats were used for establishing the LPS-induced depressive rat model. LPS (L-2880, Sigma, USA) dissolved in sterile 0.9% saline was intra-peritoneally injected into the Wistar rats at a dosage of 1 mg kg⁻¹. Behavioral tests were performed 24 hr after the injection for LPS. The WKY and LPS-induced depressive rats were screened using the SPT, and those exhibiting significantly reduced sucrose preference were selected for electrode implantation. Unless otherwise specified, rats with GF electrodes successfully implanted into LHb were selected and used in the behavioral tests and DBS-fMRI studies.

### Electrode implantation

MRI compatible bipolar stimulating electrodes were fabricated from GFs of 75 μm diameter as previously reported (*Zhao et al., 2020b*). For surgery, the rats were anesthetized using constant 2–2.5% isoflurane in medical-grade oxygen. Rats were secured in a stereotactic apparatus (Lab Standard Stereotaxic Instrument, Stoelting, USA) throughout the procedure. In a typical implantation, a bipolar GF microelectrode was implanted in the right LHb (AP: –3.6 mm, ML: –0.8 mm, and DV: 4.4–4.6 mm from dura) of a depressive rat. Craniotomy was sealed with a silicone elastomer (World Precision Instruments, USA). Super-Bond C&B (Sun Medical Co. Ltd., Japan), together with dental

methacrylate were used to fix the custom-made MRI compatible female header connector made of high-purity copper and electrode set onto the rat skull. Electrode tip placements within the LHb were verified for each subject by T2-weighted RARE anatomical MRI acquired immediately after implantation and H&E staining of the coronal brain sections at the end of the study. Unless otherwise specified, animals with electrode tips placed outside of the target LHb were discarded from the study and excluded from all further experimental analyses. They were also not included in the final subject numbers.

## DBS and behavioral tests

A stimulator (Model 2100, A-M Systems, USA) was used to deliver continuous electrical pulses. The constant-current pulses (biphasic and symmetric) with 300 μA amplitude and 90 μs duration at 130 Hz were used in all behavioral tests. The stimulation frequency and pulse duration were consistent with those used in clinical settings (*Dandekar et al., 2018*; *Zhou et al., 2018*).

For all the behavioral tests, at least a week was left between two consecutive tests. For SPT (*Liu et al., 2018*; *Zeldetz et al., 2018*), after being adapted with 1% sucrose solution for 48 hr, rats were deprived of both food and water for 24 hr. SPT were carried out on rats individually immediately after deprivation. Each rat was provided with free access to two bottles containing either 1% sucrose solution or tap water in a home cage for 4 hr. The two bottles were reversed every 30 min to avoid the possible effect of position preference. The SPI was calculated using the following formula: SPI = sucrose intake/total intake. SPT upon LHb-DBS was carried out immediately after application of 15 min of LHb-DBS.

The FST was performed as previously reported (*Porsolt et al., 1977*; *Slattery and Cryan, 2012*). A transparent acrylic cylinder (20 cm diameter and 50 cm height) was filled with water (23±2°C temperature) for forced swimming. The water depth was set to prevent animals from touching the bottom with their tails or hind limbs. The day before the test, animals were habituated to the swimming apparatus for 15 min. About 24 hr after the habituation, rats were subjected to a 15 min FST test session, which was divided into "pre" (0–5 min), "during" (5–10 min), and "after" (10–15 min) LHb-DBS. The LHb-DBS was applied in the "during" (5–10 min) session. The data upon DBS presented in this work was either from the "during" or "after" session. Animal behaviors were videotaped from the side. Immobility was defined as the animals remaining floating or motionless with only movements necessary for keeping balance in the water. Immobility time was analyzed by Ethovision software (Noldus, version 10.0, Netherlands). The connector was sealed with a silicone elastomer (World Precision Instruments, USA) to avoid current leakage to water.

For OFT, the rat was placed in a box (65×65 cm in square and 40 cm high), and the position of the rat's body center was tracked using Supermaze software (Shanghai Xinruan Information Technology, China), with a digital video camera mounted directly above the arena. An electrical commutator and pulley system were used to allow the rat to move and turn freely within the box. The rat was subjected to a 15 min OFT test session, which was divided into "pre" (0–5 min), "during" (5–10 min), and "after" (10–15 min) LHb-DBS, with locomotor performance recorded. The LHb-DBS was applied in the "during" (5–10 min) session. The data upon DBS presented in this work was either from the "during" or "after" session. This selection was remained consistent for OFT and FST data for each individual rat. Several behavioral indexes, including average speed, duration of immobility, traveling distance in total area, numbers of entries, time spent within, average speed, and total distance traveled in the center area of the OFT, were recorded and analyzed with Supermaze software.

## MRI acquisitions

All MRI experiments were performed in a Bruker 9.4T scanner with Bruker's 86 mm volume coil for transmission and a 2 cm diameter single-loop surface coil for receiving (ParaVision version 6.0.1 for MRI acquisitions). Rats were anesthetized with 4% isoflurane, followed by a bolus injection of dexmedetomidine (0.022 mg kg$^{-1}$). During MRI scanning, isoflurane (0.5%) delivered via a nose cone combined with continuous infusion of dexmedetomidine (0.015 mg kg$^{-1}$ hr$^{-1}$) was used to maintain anesthesia (*Brynildsen et al., 2017*). Animal temperature, respiration, and blood oxygen saturation were all monitored and within normal ranges (Model 1025, SA Instruments, USA). Body temperature was maintained at 37 ± 0.5°C using a circulated hot water bed and a hot air blower.

T2-weighted anatomical images were acquired using RARE sequence with the following parameters: TR/TE = 2500/33 ms, RARE factor = 8, Field of View (FOV) = 30 × 30 mm², matrix = 256 × 256, slice thickness = 0.8 mm, and contiguous 20 slices without gap in the axial direction. All fMRI data were acquired using a two-shot gradient-echo echo-planar imaging (EPI) sequence with the following parameters: TR/TE = 500/13 ms, FOV = 30 × 30 mm², matrix = 80 × 80, flip angle = 55°, repetitions = 210, slice thickness = 0.8 mm, and contiguous 18 slices without gap in the axial direction.

The fMRI scans were acquired on anesthetized rats for 210 s (210 repetitions), during which stimulation was applied in a 60 s-OFF/30 s-ON/120 s-OFF cycle, with the following parameters: bipolar square-wave current with an amplitude of 600 µA, frequency of 130 Hz, and pulse width of 90 µs. A higher current amplitude at 600 µA was used in DBS-fMRI studies to circumvent the issue of low DBS-fMRI sensitivity at low stimulus amplitude. This generated BOLD response patterns that were qualitatively similar yet more robust than 300 µA. The EPI scans were repeated three times per rat for within subject averaging.

## fMRI data analysis

The fMRI data analysis was performed using a custom-written code developed using Matlab (R2018a, MathWorks, USA) and SPM12 (http://www.fil.ion.ucl.ac.uk/). The preprocessing included the following steps. (1) EPI images were converted from Bruker format to NIFTI format (nominal voxel size was enlarged 10-fold to facilitate image processing in Statistical Parametric Mapping (SPM)). (2) After image format conversion, the rat brain was extracted manually from each image using ITK-SNAP (http://www.itksnap.org/). (3) The images of each scan were corrected for slice timing and realigned to the first volume for that scan. (4) For spatial normalization, the realigned EPI images were co-registered to the rat's own T2 anatomical images, which were normalized to a rat brain template. (5) Spatial smoothing was applied with a 0.8 mm full width at half maximum Gaussian kernel. After preprocessing, statistical analysis was conducted across subjects using a general linear model with reference to the stimulation paradigm, and a modified rodent hemodynamic response function was used (*Chen et al., 2020*). Standard first-level analysis was performed for each EPI scan of each animal, and the corresponding *t*-statistic maps of each subject were calculated, with false discovery rate-corrected set at p<0.001 and cluster size set at >10 voxels.

For time course analysis on ROIs, 11 ROIs were anatomically defined using group averaged activation maps and after applied to co-registered data. The 11 ROIs included the septum, DBN, LPO, LHA, IPN/VTA, DRN, mPFC, Cg, RS, LHb, and SLEA. Since the IPN is close to the VTA and it is difficult to accurately distinguish these two areas in fMRI studies, we labeled this region as IPN/VTA. For each scan, the time series was converted to relative BOLD response ΔS(t)/S0, where ΔS(t) was generated by subtracting the mean of pre-stimulation period (S0) of that scan. The BOLD signal time courses were calculated for each ROI. Moreover, the mean statistics beta values of these ROIs were also calculated for each subject.

## H&E staining

Electrode tip localization within LHb was confirmed by H&E staining at the end of the study. Rats were anesthetized and transcardially perfused with 100–150 mL PBS, followed by 200–300 mL 4% paraformaldehyde (PFA) in PBS. The brain was taken out with the implants removed from the skull. After being immersed in 4% PFA solution for 24 hr at room temperature for the purpose of post-fixation, the brain was immersed successively in 10 (w/v), 20 (w/v), and 30% (w/v) sucrose solution for dehydration until it settled to the bottom of the solution. The brain was segmented coronally to remove extra brain tissue, leaving out ~3 mm thick section which included the entire habenula. The tissue sample was cryoprotected using the optimal cutting temperature (OCT) compound (Tissue-Tek, USA) and frozen at –80°C to stiffen the OCT compound. Frozen tissues were sliced coronally into 7–10 µm thick sections using a cryostat machine (Leica CM3050 S, Germany), mounted on glass slides, and stored at –20°C.

The frozen brain coronal slices were put at room temperature for 15 min and then rinsed with PBS for 5 min to remove OCT compound. The cell nuclei were stained with hematoxylin solution (#BA4097, Baso, China) for 1–2 min and rinsed with water. Then the slices were put into 1% hydrochloric acid ethanol solution for differentiation for 1–5 s and rinsed with water. The differentiation was followed by a 50°C water rinse for 10–20 s to stop the action of the acid alcohol. The color of nucleus was checked

using microscope. The cytoplasm was stained with eosin solution (#BA4022, Baso, China) for 1–2 min and rinsed with water at room temperature. The slices were then dehydrated using graded alcohol in ascending order as 70, 90, and 95% ethanol. The slices were placed in each of these alcohol solutions for 5–10 s and finally rinsed with absolute ethanol twice for a total of 4 min. After being placed in xylene solution to make cells transparent, the slices were mounted on coverslips and sealed with neutral balsam mounting medium (Zsbio, China). The images of the H&E-stained slices were captured with a 4× objective lens on an Olympus microscope (Olympus BX51, Japan).

## Acknowledgements

We thank Ms. Shuang Liu, Prof. Gang Wang, and Prof. Jian Yang from Beijing Anding hospital, Capital medical University for assistance in behavioral analysis, Dr. Xiaobo Jia from School of Life Sciences at Peking University for assistance on tissue sectioning, and Dr. Xueting Sun from College of Future Technology at Peking University for assistance in tissue staining. X.D. acknowledged the support by grants from the National Natural Science Foundation of China (T2188101, 21972005), the STI2030-Major Projects (2021ZD0202204, 2021ZD0202200), National Key R&D Program of China (2021YFF1200700), and Natural Science Foundation of Beijing Municipality (JQ20008). Z.L. acknowledged the support by grants from the National Science and Technology Innovation 2030 Major Program (2021ZD0200100), Strategic Priority Research Program of Chinese Academy of Sciences (XDBS01030100), Shanghai Municipal Science and Technology Major Project (2018SHZDZX05), the National Natural Science Foundation of China (82171899), and Lingang Laboratory (LG202104-02-06).

## Additional information

### Funding

| Funder | Grant reference number | Author |
| --- | --- | --- |
| National Natural Science Foundation of China | T2188101 | Xiaojie Duan |
| STI2030-Major Projects | 2021ZD0202204 | Xiaojie Duan |
| National Key R&D Program of China | 2021YFF1200700 | Xiaojie Duan |
| Natural Science Foundation of Beijing Municipality | JQ20008 | Xiaojie Duan |
| National Science and Technology Innovation 2030 Major Program | 2021ZD0200100 | Zhifeng Liang |
| Strategic Priority Research Program of Chinese Academy of Sciences | XDBS01030100 | Zhifeng Liang |
| Shanghai Municipal Science and Technology Major Project | 2018SHZDZX05 | Zhifeng Liang |
| National Natural Science Foundation of China | 82171899 | Zhifeng Liang |
| Lingang Laboratory | LG202104-02-06 | Zhifeng Liang |
| National Natural Science Foundation of China | 21972005 | Xiaojie Duan |
| STI2030-Major Projects | 2021ZD0202200 | Xiaojie Duan |

The funders had no role in study design, data collection and interpretation, or the decision to submit the work for publication.

## Author contributions
Gen Li, Conceptualization, Resources, Data curation, Software, Formal analysis, Validation, Investigation, Visualization, Methodology, Writing - original draft, Writing - review and editing; Binshi Bo, Data curation, Software, Formal analysis, Visualization, Methodology, Writing - review and editing; Puxin Wang, Peixing Qian, Formal analysis, Validation, Investigation, Writing - review and editing; Mingzhe Li, Yuyan Li, Investigation, Writing - review and editing; Chuanjun Tong, Software, Formal analysis, Investigation, Writing - review and editing; Kaiwei Zhang, Resources, Software, Formal analysis; Baogui Zhang, Validation, Investigation, Methodology, Writing - review and editing; Tianzi Jiang, Resources, Methodology, Writing - review and editing; Zhifeng Liang, Resources, Software, Supervision, Validation, Investigation, Methodology, Writing - review and editing; Xiaojie Duan, Conceptualization, Resources, Data curation, Formal analysis, Supervision, Funding acquisition, Validation, Investigation, Writing - original draft, Project administration, Writing - review and editing

## Author ORCIDs
Gen Li ⬤ http://orcid.org/0000-0002-7034-1893
Tianzi Jiang ⬤ http://orcid.org/0000-0001-9531-291X
Xiaojie Duan ⬤ http://orcid.org/0000-0001-7799-3897

## Ethics
Our procedures for handling the animals complied with the Beijing Administration Rules of Laboratory Animals and the National Standards of Laboratory Animal Requirements of Environment and Housing Facilities (GB 14925-2010), and were approved by the Institutional Animal Care and Use Committee of Peking University (#FT-DuanXJ-1).

## Decision letter and Author response
Decision letter https://doi.org/10.7554/eLife.84693.sa1
Author response https://doi.org/10.7554/eLife.84693.sa2

# Additional files

## Supplementary files
• MDAR checklist

• Supplementary file 1. Statistics summary for *Figure 2* (Supplementary file 1a), *Figure 2—figure supplement 1* (Supplementary file 1a), and *Figure 5* (Supplementary file 1b).

## Data availability
All data generated or analyzed during this study are included in the manuscript and supplementary information. Source data files for behavioral tests and fMRI studies, as well as codes used for MRI analysis, have been provided on the Open Science Framework under ID 48f5m (https://osf.io/48f5m/). Raw videos of the animals can be made available upon reasonable request and subsequent approval from the relevant departments, including the Institutional Animal Care and Use Committee and Communication Office of Peking University, as well as of the Institute of Neuroscience, Chinese Academy of Sciences. To request access to the animal videos, interested parties should contact the corresponding authors, who will provide guidelines and policies for getting the approval.

The following dataset was generated:

| Author(s) | Year | Dataset title | Dataset URL | Database and Identifier |
|---|---|---|---|---|
| Li G, Bo B, Wang P, Qian P, Li M, Li Y, Tong C, Zhang K, Zhang B, Jiang T, Liang Z, Duan X | 2023 | Instantaneous antidepressant effect of lateral habenula deep brain stimulation in rats studied with functional magnetic resonance imaging | https://osf.io/48f5m/ | Open Science Framework, 48f5m |

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
