## [Editor Report]

This important paper is a real tour de force that combines functional MRI, behaviour, and brain stimulation to characterise the effect of stimulation of the lateral habenula in a rodent model for depression. The results are compelling and provide additional information potentially relevant to both surgical targeting and mechanism of action for this deep brain stimulation target.

---

## [Decision Letter]

**Decision letter after peer review:**

Thank you for submitting your article "Instantaneous antidepressant effect of lateral habenula deep brain stimulation in rats studied with functional magnetic resonance imaging" for consideration by *eLife*. Your article has been reviewed by 2 peer reviewers, including Saad Jbabdi as the Reviewing Editor and Reviewer #2, and the evaluation has been overseen by and Christian Büchel as the Senior Editor. The following individual involved in review of your submission has agreed to reveal their identity: John Younce (Reviewer #1).

*Reviewer #1 (Recommendations for the authors):*

– I would mark the controls as "control" rather than "SD" for readability in Figure 1.

– Figures 3 and 4 would benefit from a more clear organization of brain regions used as seeds (e.g. limbic regions together, cortical regions together, etc).

– Figure 4 needs to be better organized between behavioural categories for readability.

– While the groupings provided in Figure 5 appear reasonable to support the hypothesis that medial placements are associated with more robust response, a color-coded scatterplot of placement laterality vs each behavioural response would help with transparency here. This could be added to supplementary material if there is not enough room in the figure.

---

## [Author Response]

Reviewer #1 (Recommendations for the authors):– I would mark the controls as "control" rather than "SD" for readability in Figure 1.

We thank the reviewer for the advice. We have marked the controls as “Control” in Figure 2 and Figure 2—figure supplement 1 as suggested by the reviewer.

– Figures 3 and 4 would benefit from a more clear organization of brain regions used as seeds (e.g. limbic regions together, cortical regions together, etc).

We thank the reviewer for the advice. We have reorganized the data of different brain regions in the order of limbic regions, dopaminergic regions, serotonergic regions, cortical regions and others (LHb and SLEA) in Figure 3, Figure 4, Figure 5, and Figure 4-figure supplement 1, as suggested by the reviewer.

– Figure 4 needs to be better organized between behavioural categories for readability.

We thank the reviewer for the advice. We have reorganized the data in Figure 4 according to different behavioural indicators to make them more readable. Figure 4a is the correlation analysis on change of average speed, and Figure 4b is the correlation analysis on number of entries into centre.

– While the groupings provided in Figure 5 appear reasonable to support the hypothesis that medial placements are associated with more robust response, a color-coded scatterplot of placement laterality vs each behavioural response would help with transparency here. This could be added to supplementary material if there is not enough room in the figure.

We thank the reviewer for the advice. We have provided the color-coded scatter plots of electrode tips placement *versus* different behavioural response indicators as Figure 5-figure supplement 2 in the revised manuscript, which show the stimulation position-dependent differences on behavioural responses.